# VideoCap-R1: Enhancing MLLMs for Video Captioning via Structured Thinking

## Abstract

While recent advances in reinforcement learning have significantly enhanced reasoning capabilities in large language models (LLMs), these techniques remain underexplored in multi-modal LLMs for video captioning. This paper presents the first systematic investigation of GRPO-based RL post-training for video MLLMs, with the goal of enhancing video MLLMs' capability of describing actions in videos. Specifically, we develop the **VideoCap-R1**, which is prompted to first perform structured thinking that analyzes video subjects with their attributes and actions before generating complete captions, supported by two specialized reward mechanisms: a LLM-free think scorer evaluating the structured thinking quality and a LLM-assisted caption scorer assessing the output quality. The RL training framework effectively establishes the connection between structured reasoning and comprehensive description generation, enabling the model to produce captions with more accurate actions. Our experiments demonstrate that VideoCap-R1 achieves substantial improvements over the Qwen2VL-7B baseline using limited samples (1.5k) across multiple video caption benchmarks (DREAM-1K: **+4.4** event F1, VDC: **+4.2** Acc, CAREBENCH: **+3.1** action F1, **+6.9** object F1) while consistently outperforming the SFT-trained counterparts, confirming GRPO's superiority in enhancing MLLMs' captioning capabilities.

## 1 Introduction

Test-time scaling has been proven to effectively enhance the reasoning capabilities of large language models (LLMs), as demonstrated by OpenAI's o1 (Jaech et al., 2024), Deepseek-R1 (Guo et al., 2025), and Kimi-1.5 (Team et al., 2025), which exhibit strong performance in complex logical tasks such as mathematics(Lightman et al., 2023) and coding(Jain et al., 2024). Notably, Deepseek-R1 showcases the potential of LLMs to develop reasoning abilities without any supervised data, relying solely on pure reinforcement learning with rule-based verifiable rewards.

Many researchers(Meng et al., 2025; Liu et al., 2025; Shen et al., 2025; Feng et al., 2025; Li et al., 2025; Zhou et al., 2025) have devoted significant efforts to extending Deepseek-R1's paradigm to the multimodal large language models (MLLMs), aiming to improve visual reasoning capabilities. These works collectively validate the superiority of the GRPO(Shao et al., 2024) algorithm over supervised fine-tuning (SFT) in specific visual tasks, such as visual question answering(Zhao et al., 2025b; Yang et al., 2024c), spatial grounding(Yu et al., 2016), and temporal grounding(Gao et al., 2017). However, they primarily focus on verifiable problems (e.g., math(Lu et al., 2023; Zhang et al., 2024a; He et al., 2024), multiple-choice questions(Zhao et al., 2025b; Yang et al., 2024c; Hu et al., 2025)), leaving open-ended problems like video captioning(Chai et al., 2025; Xu et al., 2024b; Wang et al., 2024a; Caba Heilbron et al., 2015) underexplored.

In video captioning, most existing approaches rely on manually annotated or commercial model-generated (e.g., GPT-4o(Hurst et al., 2024), Gemini-2.0(Pichai et al., 2024)) high-quality video description datasets for instruction tuning, which is both time-consuming and costly. Inspired by the Chain-of-Thought (CoT) paradigm(Xu et al., 2024a), we decompose the captioning task into two sequential steps: first prompting the model to perform structured reasoning that analyzes video subjects, their attributes and actions, then requiring it to synthesize these elements into a complete caption.

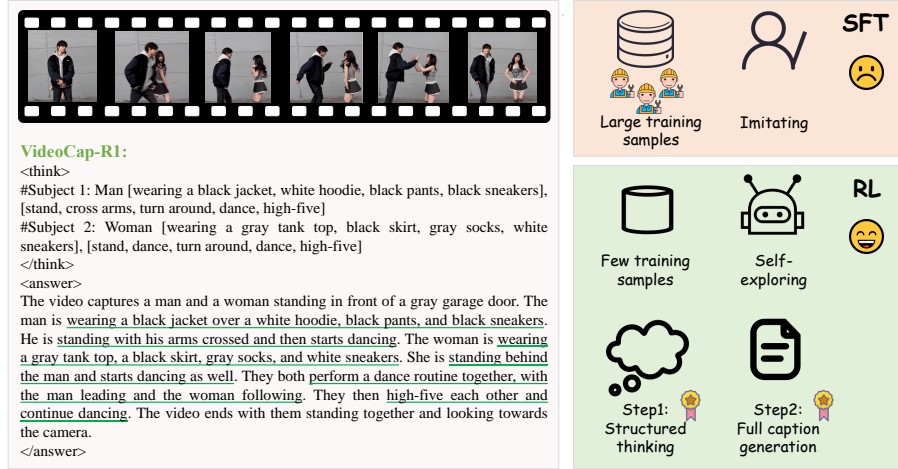

Figure 1: **Motivation of VideoCap-R1.** SFT requires costly high-quality data and the trained model merely imitates training distributions. VideoCap-R1 instead decomposes captioning into structured thinking and answering phases, optimized via GRPO with dual rewards for thinking and caption. By effectively establishing the connection between structured reasoning and comprehensive description generation, VideoCap-R1 can generate captions with more accurate actions.

We initially constructed an instruction-tuning dataset incorporating this structured reasoning process and performed supervised fine-tuning (SFT) on the baseline model. However, we observed that the model only learned superficial reasoning patterns - it acquired the output format without establishing meaningful connections between the structured reasoning process and final descriptions. We attribute this limitation to the train-inference discrepancy in SFT: during training, the model generates tokens conditioned on ground-truth prefixes, whereas during inference it must rely on its own predictions, resulting in poorer performance when generating captions with structured thinking process compared to direct generation.

To address these limitations, we leverage the recent success of GRPO-based RL post-training strategies, which can provide online rewards for correct reasoning paths. This enables the model to genuinely learn the two-step process of first solving simpler subproblems before generating complete captions. The primary difficulty in applying GRPO to video captioning lies in effectively rewarding free-form textual outputs. To overcome this, we designed two specialized reward mechanisms: a LLM-free think scorer that evaluates reasoning quality, and a LLM-assisted caption scorer that assesses output quality. Based on this framework, we developed VideoCap-R1, which first identifies key visual elements before generating detailed descriptions, significantly enhancing the baseline model's captioning capability even with limited training samples(1.5k).

Our main contributions are summarized as follows:

- We propose a novel structured reasoning process specifically designed for caption generation, where the model first identifies key visual subjects along with their attributes and actions before generating comprehensive descriptions. Under GRPO training, this structured approach demonstrates substantial gains in caption quality.

- We present the first successful application of GRPO to open-ended video captioning tasks. Our work introduces two meticulously designed reward mechanisms that jointly assess both the reasoning process and the final caption quality. Based on this framework, we develop VideoCap-R1, which demonstrates consistent performance improvements over baseline model Qwen2-VL-7B(Wang et al., 2024b) even with limited training samples(1.5k) across three challenging benchmarks: DREAM-1K(**+4.4** event F1), VDC(**+4.2** accuracy), and CAREBENCH(**+3.1** action F1, **+6.9** object F1).

- Our analysis reveals that SFT only enables models to learn superficial reasoning patterns. This is evidenced by models fine-tuned with structured thinking augmented data under-

performing those trained on standard captioning data. In contrast, our GRPO-based RL approach enables the model to develop genuinely beneficial reasoning patterns, outperforming SFT-based counterparts when using identical training datasets, regardless of structured thinking augmentation.

## 2 RELATED WORK

**Video caption models.** Video captioning is one of the most fundamental tasks in video understanding. Since video captioning datasets are commonly used in the pre-training phase of multimodal large language models (MLLMs) to align linguistic and visual space, general video MLLMs(Zhang et al., 2024c; Wang et al., 2024b; Chen et al., 2024c) typically possess basic video captioning capabilities. The prevailing approach to enhancing MLLMs' video captioning performance involves constructing high-quality video description datasets for instruction tuning. For instance, ShareGPT4Video(Chen et al., 2024b) designs a differential video captioning strategy, leveraging GPT-4V(Achiam et al., 2023) to annotate videos and develop ShareCaptioner-Video. Similarly, Shot2Story (Han et al., 2023) and Vript(Yang et al., 2024b) employ GPT-4V for video captioning. LLaVA-video (Zhang et al., 2024c) introduces a recurrent detailed caption generation pipeline powered by GPT-4o, enabling fine-grained descriptions for videos of arbitrary length. Tarsier2(Yuan et al., 2025) further advances this direction by curating 40 million large-scale video-text pairs for pretraining and 150K human-annotated video descriptions with temporal grounding for instruction tuning. While these specialized video description models (VDCs)(Chen et al., 2024b; Zhang et al., 2024c; Yuan et al., 2025; Han et al., 2023) excel at generating detailed captions, they predominantly rely on large-scale, manually annotated instruction-tuning datasets, which are costly and time-consuming to produce. In contrast, our work explores training efficiency by leveraging reinforcement learning (RL) to guide the model in reasoning before generating captions. Under the same data budget, our approach outperforms supervised fine-tuning (SFT), demonstrating superior data efficiency.

**Reinforcement learning for MLLMs.** Reinforcement learning (RL) is typically applied during the post-training phase of LLMs and has been proven to be critical for mitigating hallucination or enhancing reasoning capabilities. The OpenAI's o1 model(Jaech et al., 2024) first demonstrated the significant potential of test-time scaling in improving model reasoning. Subsequently, Deepseek-R1(Guo et al., 2025) showed that reinforcement learning with rule-based verifiable rewards could effectively enhance LLMs' performance in mathematical and coding tasks. This approach inspired numerous efforts to extend the R1's paradigm to multimodal domains to improve MLLMs' reasoning abilities. For instance, MM-EUREKA(Meng et al., 2025) focuses on multimodal mathematical tasks with visual inputs, revealing a "visual aha moment" where the model reaffirms its answer by re-perceiving the image. Furthermore, Visual-RFT(Liu et al., 2025) and VLM-R1(Shen et al., 2025) enhance MLLMs' performance in fundamental visual perception tasks, including detection and grounding. In the video domain, prior work(Wang et al., 2025; Zhao et al., 2025a; Li et al., 2025) has explored the effectiveness of GRPO in tasks such as temporal grounding, sentiment analysis, object tracking, and general visual question answering. However, open-ended tasks like video captioning remain understudied. For instance, VideoChat-R1 (Li et al., 2025)attempted to improve video description quality using event recall as a reward function, but the generated captions remained far from satisfactory. Our work addresses this gap by systematically designing and evaluating reward functions tailored for captioning, successfully adapting GRPO to this task and significantly improving description quality.

## 3 METHODOLOGY

### 3.1 PRELIMINARY

#### 3.1.1 GROUP RELATIVE POLICY OPTIMIZATION

Group Relative Policy Optimization (GRPO)(Shao et al., 2024) is an enhanced variant of Proximal Policy Optimization (PPO)(Schulman et al., 2017). GRPO obviates the need for additional value function and uses the average reward of multiple sampled outputs for the same question to estimate the advantage. To be specific, for each question-answer pair $(q, a)$, the old policy $\pi_{\theta_{old}}$ samples a group of outputs $\{o_1, o_2, \ldots, o_G\}$ and a predefined reward function is used to evaluate these outputs

to get their corresponding rewards $\{r_1, r_2, \ldots, r_G\}$. Then the advantage of the $i$-th response relative to other sampled responses is calculated by normalizing the group-level rewards $\{r_1, r_2, \ldots, r_G\}$: $\hat{A}_i = \frac{r_i - \text{mean}(\{r_1, r_2, \ldots, r_G\})}{\text{std}(\{r_1, r_2, \ldots, r_G\})}$. GRPO encourages the model to prioritize the responses with higher advantages within the group by updating the policy $\pi_\theta$ using the following clipped surrogate objective:

$$
\mathcal{J}_{\text{GRPO}}(\theta) = \mathbb{E}_{(q,a) \sim \mathcal{D}, \{o_i\}_{i=1}^G \sim \pi_{\theta_{\text{old}}}(\cdot|q)}
$$
$$
\left[ \frac{1}{G} \sum_{i=1}^G \frac{1}{|o_i|} \sum_{t=1}^{|o_i|} \left( \min\left( \frac{\pi_\theta(o_{i,t} \mid q, o_{i,<t})}{\pi_{\theta_{\text{old}}}(o_{i,t} \mid q, o_{i,<t})} \hat{A}_i, \text{clip}\left( \frac{\pi_\theta(o_{i,t} \mid q, o_{i,<t})}{\pi_{\theta_{\text{old}}}(o_{i,t} \mid q, o_{i,<t})}, 1-\varepsilon, 1+\varepsilon \right) \hat{A}_i \right) - \beta D_{\text{KL}}(\pi_\theta || \pi_{\text{ref}}) \right) \right], \tag{1}
$$

where $\varepsilon$ and $\beta$ are hyper-parameters. $\varepsilon$ is the clipping range of importance sampling ratio and KL divergence is adopted to regularize the policy model, preventing excessive deviation from the reference model.

## 3.2 CAPTION REWARD MODELING

The reward function plays a pivotal role in determining the optimization direction of reinforcement learning. Deepseek-R1 adopts a rule-based reward system comprising two primary components: format reward and accuracy reward. Building upon Deepseek-R1's framework, we introduce novel caption-specific reward functions to guide policy optimization, whose components and implementation details are elaborated as follows.

### 3.2.1 TWO-STEP CAPTION GENERATION STRATEGY

While video captioning is fundamentally a perceptual task requiring comprehensive description of visual elements, it presents unique challenges compared to visual question answering. Unlike VQA tasks(Li et al., 2024b; Fu et al., 2024) that focus only on question-relevant content, video captioning demands complete coverage of all significant elements within potentially complex video sequences.

Inspired by the Chain-of-Thought (CoT) paradigm(Xu et al., 2024a; Zhang et al., 2024b; Thawakar et al., 2025; Yao et al., 2024) that decomposes complex tasks into manageable sub-problems, we propose a two-step caption generation strategy. Our approach first requires the model to perform structured reasoning that analyzes and identifies key video subjects along with their attributes and actions in the thinking process and then synthesize these elements into coherent captions in the final outputs, as illustrated in Figure 1. The training prompt is detailed in the Appendix F. We employ format reward as Deepseek-R1 to ensure the model adheres to this format. The two-step caption generation strategy mirrors compositional writing, where one first outlines key points before developing complete paragraphs. Our experimental results confirm that this explicit reasoning framework significantly enhances the model's capability to describe actions and events in videos.

### 3.2.2 LLM-FREE THINK SCORER

To effectively reward accurate identification of key elements, we implement a LLM-free think scorer for the intermediate thinking stage. We extract subject names, attribute lists, and action lists through regular expression matching and compute corresponding precision and recall metrics against ground truth annotations.

Formally, let the model predict $N$ entities, each containing a name $\text{name}_I^p$, an attribute list $\text{attr}_I^p$, and an action list $\text{act}_I^p$ ($1 \le I \le N$), while the ground truth contains $M$ entities with corresponding $\text{name}_J^g$, $\text{attr}_J^g$, and $\text{act}_J^g$ ($1 \le J \le M$). For each predicted action list $\text{act}_I^p = \{p_i\}_{i=1}^n$ and its corresponding ground truth action list $\text{act}_J^g = \{g_j\}_{j=1}^m$, we formulate a bipartite graph matching problem where nodes represent predicted and ground truth actions respectively, with edge weights $\text{sim}(p_i, g_j)$ computed as the dot product of their word embeddings encoded by M3-Embedding (Chen et al., 2024a). By computing the dot product between each action embedding from $\text{act}_I^p$ and those from $\text{act}_J^g$, we obtain their similarity matrix $\text{SIM}(\text{act}_I^p, \text{act}_J^g) \in \mathbb{R}^{n \times m}$. To avoid matching dissimilar actions, we apply a similarity threshold $\delta$, setting edge weights below it to 0. Our goal is to find the optimal one-to-one assignment $\hat{A}$ that maximizes total similarity:

$$
\hat{A} = \arg\max_{A \in \Omega} \sum_{(i,j) \in A} \text{SIM}(\text{act}_I^p, \text{act}_J^g)_{i,j}, \tag{2}
$$

where $\Omega$ represents the set of all valid assignments between predictions and ground truths. We solve this matching problem using the Jonker-Volgenant algorithm (Jonker & Volgenant, 1988) and subsequently define the precision and recall score for the action sequence as follows:

$$P(\text{act}_I^p, \text{act}_J^g) = \frac{1}{n} \sum_{(i,j) \in \hat{A}} \text{SIM}(\text{act}_I^p, \text{act}_J^g)_{i,j}, \quad R(\text{act}_I^p, \text{act}_J^g) = \frac{1}{m} \sum_{(i,j) \in \hat{A}} \text{SIM}(\text{act}_I^p, \text{act}_J^g)_{i,j}. \quad (3)$$

The F1 score can be calculated as $F1(\text{act}_I^p, \text{act}_J^g) = \frac{2PR}{P+R}$. The precision, recall, and F1 score for attribute lists are calculated in the same manner.

Since videos may contain multiple objects, we first establish one-to-one correspondences at the entity level between predicted and ground truth objects before computing attribute and action F1 scores. We define the similarity between the $I$-th predicted entity and $J$-th ground truth entity as:

$$\text{sim}(p_I, g_J) = F1(\text{attr}_I^p, \text{attr}_J^g) + F1(\text{act}_I^p, \text{act}_J^g) + \text{sim}(\text{name}_I^p, \text{name}_J^g). \quad (4)$$

Using the same matching algorithm, we obtain the optimal entity-level assignment $\hat{A}$. The overall metrics for action sequences are then calculated as:

$$P_{\text{overall\_act}} = \frac{1}{N} \sum_{(I,J) \in \hat{A}} P(\text{act}_I^p, \text{act}_J^g), \quad R_{\text{overall\_act}} = \frac{1}{M} \sum_{(I,J) \in \hat{A}} R(\text{act}_I^p, \text{act}_J^g). \quad (5)$$

The F1 score can be calculated as $F1_{\text{overall\_act}} = \frac{2P_{\text{overall\_act}} R_{\text{overall\_act}}}{P_{\text{overall\_act}} + R_{\text{overall\_act}}}$. The overall precision, recall, and F1 score for attribute lists are computed in the same manner. The final thinking score(Tscore) for the reasoning process combines these metrics with weighted coefficients:

$$\text{Tscore} = 0.6 \times F1_{\text{overall\_act}} + 0.4 \times F1_{\text{overall\_attr}}. \quad (6)$$

### 3.2.3 LLM-Assistant Caption Scorer

As the saying goes, "a picture is worth a thousand words", and a video can be described in numerous valid ways. This makes direct comparison between predicted and ground truth captions challenging for scoring. We therefore design multiple scoring dimensions for caption evaluation, ultimately combining them into an overall score. We employ Qwen2.5-72B(Yang et al., 2024a) as our judge model due to its exceptional language understanding capabilities. Our investigation explores two distinct caption scoring approaches: (1) direct rule-based scoring by the LLM, and (2) event coverage computation through LLM-assisted event extraction, detailed as follows:

COMPLETENESS-NATURALNESS SCORE (CNSCORE)

Since the model first identifies key entities and their attributes/actions during reasoning, the final caption should naturally organize these elements. We evaluate this through two metrics:

$$\text{CNscore} = \frac{\text{Completeness}_{\text{score}} + \text{Naturalness}_{\text{score}}}{20}, \quad (7)$$

where $\text{Completeness}_{\text{score}} \in [0, 10]$ measures coverage of reasoned elements, and $\text{Naturalness}_{\text{score}} \in [0, 10]$ assesses linguistic fluency and human-like description quality. The scoring prompt for Qwen2.5-72B is provided in Appendix E.

EVENT SCORE (ESCORE)

Naturalness scoring exhibits significant subjectivity and is susceptible to the inherent biases of the judge model, potentially leading to reward hacking(Gao et al., 2023) where the model optimizes for generating captions that artificially inflate judge scores while substantially deviating from the desired

caption quality objectives. To mitigate this, we avoid direct scoring by the judge model. Considering video descriptions comprise sequences of events (who did what), we evaluate the predicted caption based on event coverage:

$$
\text{Escore} = \begin{cases} 0 & \text{if event\_coverage} < \delta_1, \\ 0.5 & \text{if } \delta_1 \leq \text{event\_coverage} < \delta_2, \\ 1 & \text{if event\_coverage} \geq \delta_2. \end{cases} \tag{8}
$$

Here, event coverage represents the proportion of ground truth events entailed by the predicted caption, and we employ Qwen2.5-72B as the judge model to determine these entailment relationships. The prompt for Qwen2.5-72B is the same as Tarsier(Wang et al., 2024a).

### 3.3 ENHANCING VIDEO DESCRIPTION CAPABILITIES OF VIDEO MLLMS VIA GRPO

**Reward Function.** The final reward function for GRPO-based training combines multiple scoring components:Reward = Format_score + Tscore + Escore.

**Training Data Construction.** To effectively reward the model's reasoning process, we construct specialized training data containing explicit structured reasoning annotations. Rather than randomly sampling from existing video captioning datasets, we developed a systematic data selection and annotation pipeline to curate videos exhibiting dynamic motions while ensuring the final training set maintains: (1) diverse action categories with balanced distribution across the dataset, and (2) comprehensive annotations that include both final captions and corresponding reasoning process. The complete data curation pipeline is detailed in the Appendix C. Through this process, we established a carefully annotated dataset comprising 1.5k training samples for our experiments. Surprisingly, our model demonstrates substantial performance gains in video captioning despite the limited training set size, validating both the efficacy of our data curation strategy and the robustness of the proposed algorithm. In future work, we plan to scale up training with more data to further boost performance.

## 4 EXPERIMENTS

### 4.1 EXPERIMENT SETUPS

**Implementation Details.** We employ Qwen2-VL-7B-Instruct (Wang et al., 2024b) as our baseline model. For both supervised fine-tuning (SFT) and reinforcement learning (RL) training, we utilize the Swift framework(Zhao et al., 2024), and we uniformly sample up to 32 frames for each video and resize each frame to a maximum of 460,800 pixels. All experiments are conducted on 8 H800-80GB GPUs. More implementation details are provided in Appendix B.

**Evaluation Benchmarks.** We evaluate our model on three video captioning benchmarks: DREAM-1K (Wang et al., 2024a), VDC (Chai et al., 2025), and CAREBENCH (Xu et al., 2024b). DREAM-1K is specifically designed to assess fine-grained action and event description capabilities, featuring dynamic and diverse video content with human-written reference captions. The VDC benchmark comprises over 1,000 videos with exceptionally detailed captions, enabling rigorous evaluation of detailed video description quality. For this benchmark, we employ the official VDCSCORE metric to assess the detailed captioning subtask. CAREBENCH provides comprehensive evaluation of both static objects(spatial elements) and dynamic actions(temporal elements) in captions. To ensure fair comparison, we strictly adhere to the experimental settings specified in each benchmark.

### 4.2 MAIN RESULTS AND ANALYSIS

We conduct comprehensive evaluations of VideoCap-R1 across three established benchmarks, comparing against both general video MLLMs and specialized captioning models (Table 1). VideoCap-R1 demonstrates substantial improvements over the Qwen2-VL-7B baseline even with limited training samples(1.5k), achieving gains of **+4.4** event F1 on DREAM-1K, **+4.2** accuracy on VDC, **+3.1** action F1 and **+6.9** object F1 on CAREBENCH. Furthermore, our model outperforms all general MLLMs and specialized captioning models by significant margins on both VDC and CAREBENCH. While showing marginally lower event F1 (-0.4%) than Tarsier-7B on DREAM-1K, VideoCap-R1 exhibits

Table 1: **Evaluation results on DREAM-1K,VDC and CAREBENCH.** Cells with * are reproduced using the official code. The remaining are reported numbers from literature. We highlight the **best** results in bold and second-best results with underlining.

| Model | DREAM-1K | VDC | | CAREBENCH | |
| --- | --- | --- | --- | --- | --- |
| | Event F1/P/R | Acc. | Score | Action F1/P/R | Object F1/P/R |
| *Proprietary models* | | | | | |
| Gemini-1.5-Pro(Team et al., 2024) | 36.2/37.6/34.8 | 43.1 | 2.2 | - | - |
| GPT-4o(Hurst et al., 2024) | 39.2/43.4/35.7 | - | - | - | - |
| GPT-4o mini(Hurst et al., 2024) | - | - | - | 36.8/50.2/29.1 | 33.8/49.1/25.8 |
| *Open-source models (>10B)* | | | | | |
| LLaVA-OV-72B(Li et al., 2024a) | 33.2/35.9/30.9 | - | - | - | - |
| LLaVA-Video-72B(Zhang et al., 2024c) | 34.0/37.3/31.3 | - | - | - | - |
| InternVL2.5-78B(Chen et al., 2024c) | 28.6/35.7/23.9 | - | - | 28.2/46.4/20.3 | 30.5/39.5/24.8 |
| Qwen2-VL-72B(Wang et al., 2024b) | 33.2/37.3/29.9 | - | - | 30.5/47.1/22.6 | 24.2/51.9/15.8 |
| *Open-source general MLLMs (<10B)* | | | | | |
| LLaVA-OV-7B(Li et al., 2024a) | 31.7/34.3/29.4 | 41.2 | 2.1 | - | - |
| LLaVA-Video-7B(Zhang et al., 2024c) | 32.5/37.9/28.4 | 35.0 | 1.8 | - | - |
| InternVL2.5-8B(Chen et al., 2024c) | 27.6/34.7/22.9 | 43.0 | 2.2 | 26.0/43.2/18.6 | 29.1/38.2/23.5 |
| *Open-source specialized captioning MLLMs (<10B)* | | | | | |
| Tarsier-7B(Wang et al., 2024a) | **34.6**/40.3/30.2 | 38.3* | 2.1* | 27.1/51.1/18.4 | 31.1/46.5/23.4 |
| ShareGPT4Video-8B(Chen et al., 2024b) | 20.4/27.6/16.1* | 35.6 | 1.8 | 16.5/32.6/11.0* | 20.4/42.1/13.4* |
| Vriptor(Yang et al., 2024b) | 24.4/23.6/25.1* | 38.5 | 2.0 | 23.6/48.7/15.6* | 25.3/38.1/18.9* |
| AuroraCap-7B(Chai et al., 2025) | 20.8/24.4/18.1* | 41.3 | 2.1 | 21.5/40.3/14.7* | 26.6/34.4/21.6* |
| Qwen2-VL-7B(Wang et al., 2024b) | 29.8/33.6/26.8* | 39.6* | 2.1* | 31.3/49.5/22.9* | 27.4/50.7/18.8* |
| **VideoCap-R1(Ours)** | 34.2$(+4.4)$/33.6/34.7 | **43.8**$(+4.2)$ | **2.4**$(+0.3)$ | **34.4**$(+3.1)$/48.2/26.8 | **34.3**$(+6.9)$/50.6/25.9 |

Table 2: **Comparison between SFT and RL.** Our GRPO-based two-stage generation strategy demonstrates consistent advantages over SFT across all benchmarks.

| Model | DREAM-1K(F1) | VDC(Acc.) | CARE-Action(F1) | CARE-Object(F1) | AVG |
| --- | --- | --- | --- | --- | --- |
| Baseline | 29.8 | 39.6 | 31.3 | 27.4 | 32.0 |
| +SFT | 32.8 | 39.8 | 32.6 | 31.5 | 34.2 |
| +SFT with structured thinking | 32.2 | 40.5 | 31.4 | 26.8 | 32.7 |
| **VideoCap-R1(Ours)** | **34.2** | **43.8** | **34.4** | **34.3** | **36.7** |

superior performance on the other two benchmarks, indicating stronger generalization capabilities. Notably, VideoCap-R1 achieves a state-of-the-art object F1 score of 34.3% on CAREBENCH, surpassing even the proprietary GPT-4o-mini (33.8%), and outperforms Gemini-1.5-Pro by 0.7% on VDC. These results validate that our structured reasoning approach significantly enhances both precision and recall in describing subjects and actions, ultimately improving overall caption quality.

### 4.2.1 SUPERIORITY OF RL TO SFT

Both supervised fine-tuning (SFT) and reinforcement learning (RL) are widely adopted post-training techniques for MLLMs. We investigate their respective impacts on model generalization and reasoning capabilities using identical training data (Table 2).

The SFT-trained model shows notable gains on DREAM-1K and CAREBENCH, indicating improved action/object description capabilities. However, its generated captions lack detailed attributes and contextual information, resulting in no improvement on the more demanding VDC benchmark. When we introduce structured thinking into the SFT data (Row 3), the model exhibits degraded average performance compared to standard SFT. This suggests SFT's teacher-forcing paradigm merely encourages pattern imitation without establishing genuine reasoning-caption relationships, thereby failing to benefit from the structured thinking process.

In contrast, our GRPO-based two-stage generation strategy demonstrates consistent advantages over SFT across all benchmarks, as the RL framework's inherent self-exploration mechanism coupled with dual think-and-answer rewards enables authentic task decomposition through structured reasoning, ultimately yielding higher-quality descriptions. These experimental results demonstrate that reinforcement learning achieves superior efficacy over supervised fine-tuning for open-ended video captioning tasks, significantly enhancing both model generalization and reasoning capabilities.

### 4.3 ABLATION STUDY

To validate our caption reward framework, we conduct an ablation study by systematically disabling individual reward components (Table 3). Using either think score or caption score alone improves

Table 3: **Ablation Study on Caption Reward Modeling.** A combination of think score and caption score yields significant boost in performance.

| Model | Think Score Tscore | Caption Score CNscore | Escore | DREAM-1K Event F1 | VDC Acc. | CAREBENCH Action F1 | CAREBENCH Object F1 | AVG |
|---|---|---|---|---|---|---|---|---|
| Baseline | | | | 29.8 | 39.6 | 31.3 | 27.4 | 32.0 |
| w/o Caption Score | ✔ | | | **34.3** | 40.3 | 31.4 | 28.3 | 33.6 |
| w/o Think Score | | | ✔ | 30.6 | 43.4 | 33.1 | 33.2 | 35.1 |
| Two Score | ✔ | ✔ | | 32.5 | **46.8** | **35.2** | 31.6 | 36.5 |
| | ✔ | | ✔ | 34.2 | 43.8 | 34.4 | **34.3** | **36.7** |

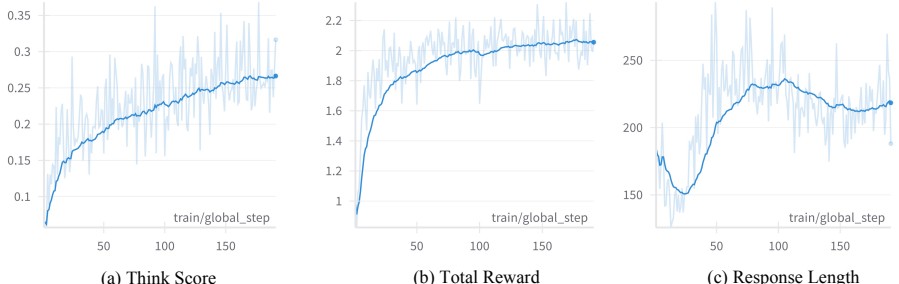

(a) Think Score      (b) Total Reward      (c) Response Length

Figure 2: The metric curves of think score, total reward and response length of **VideoCap-R1**, which show the dynamics of RL training.

model performance, while their combination yields optimal results, confirming our reward design's effectiveness. We provide further discussion on the structured thinking process in Appendix A. The bottom two rows compare our two caption scoring variants: CNscore achieves the highest performance on VDC, whereas Escore delivers the best overall performance across all three benchmarks. Notably, while CNscore leverages LLM-based direct assessment, it suffers from reward hacking - the model tends to generate psychologically nuanced descriptions that appeal to the judge (Qwen2.5-72B) but lack objective video content relevance. This phenomenon explains its suboptimal generalization. We therefore adopt Escore as our default configuration, as its event-based discriminative evaluation provides more objective scoring of factual video descriptions, ultimately producing models with superior overall capability.

### 4.4 TRAINING DYNAMICS

We primarily monitor the reinforcement learning process using three metrics: think score, total reward, and response length, as illustrated in Figure 2. The think score and total reward exhibit stable increasing trends, demonstrating that the model successfully learns to first perform structured reasoning before generating complete video descriptions, which validates the effectiveness of our carefully designed reward signals.

The response length of VideoCap-R1 initially decreases, then increases before relatively stabilizing - the early-phase pattern aligns with observations from prior work (Zeng et al., 2025). This trajectory indicates that the RL training progressively replaces the model's original reasoning patterns with the new reasoning style. Notably, the response length does not grow indefinitely but shows slight fluctuations, which we attribute to varying information density across videos. This length variation suggests the model learns to adapt its description length according to the actual video content.

### 4.5 QUALITATIVE RESULTS

Figure 3 presents qualitative comparisons between VideoCap-R1 and baseline models. Both the baseline Qwen2-VL and its SFT-finetuned version erroneously hallucinate the "stick" held by the woman as an "umbrella". In contrast, our VideoCap-R1 correctly identifies the stick while additionally providing: (1) more detailed attributes, (2) finer-grained action descriptions, and (3) even capturing subtle motions (e.g., "spins around") missed by GPT-4o. Notably, VideoCap-R1's structured reasoning

432
433
434
435
436
437
438
439
440
441
442
443
444
445
446
447
448
449
450
451
452
453
454
455
456
457
458
459
460
461
462
463
464

**Qualitative Case Study**

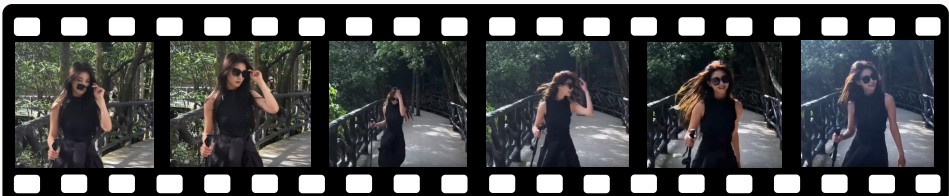

**GPT-4o:** A woman is walking along a path in a wooded area. She is holding a walking stick and occasionally adjusts her hair or sunglasses. At one point, she starts to jog or run along the path. The background consists of dense trees and foliage with sunlight filtering through.

**Qwen2-VL-7B:** The video features a woman walking down a path in a lush, green forest. She is wearing a black dress and carrying a black umbrella. As she walks, she turns around and smiles at the camera. The path is surrounded by tall trees with green leaves, creating a serene and peaceful atmosphere. The woman appears to be enjoying her walk and seems to be in a happy mood. The lighting in the video is natural, with sunlight filtering through the trees and casting shadows on the ground. Overall, the video captures a moment of tranquility and beauty in nature.

**Qwen2-VL-7B-SFT:** The video features a woman walking on a wooden path surrounded by lush greenery. She is wearing a black sleeveless top and a long black skirt, and she is holding a black umbrella in her right hand. As she walks, she turns around and smiles at the camera, then continues walking forward. The path is lined with wooden railings on both sides, and the background is filled with trees and foliage. The woman's hair is long and dark, and she appears to be enjoying her surroundings.

**VideoCap-R1:**
<think>
#Subject 1: Woman [long black hair, wearing a black sleeveless dress, sunglasses], [turn around, put on sunglasses, walk forward, spin around, adjust hair, hold walking stick, pose for the camera]
</think>
<answer>
The video captures a woman walking on a wooden path surrounded by lush greenery. She is wearing a black sleeveless dress and has long black hair. She is holding a walking stick in her right hand. The woman turns around and puts on her sunglasses, then spins around. She adjusts her hair and continues walking forward. She then spins around again and poses for the camera. The path is lined with trees and has a wooden railing on one side. The sunlight filters through the leaves, creating a dappled effect on the path. The woman appears to be enjoying her walk in the park, and the scenery is serene and peaceful.
</answer>

Figure 3: Qualitative comparison between VideoCap-R1 and baseline models. We annotate examples with: red underlines for hallucinated content, green underlines for accurate descriptions, and highlighting for correct details uniquely captured by VideoCap-R1 or Qwen2-VL-7B-SFT (vs. Qwen2-VL-7B). Our model demonstrates superior fine-grained content description capabilities.

content demonstrates strong alignment with final descriptions, evidencing that it genuinely establishes and benefits from the reasoning-description relationship, ultimately enhancing overall caption quality through this cognitive process. See Appendix D for more comparisons.

## 5 CONCLUSION AND FUTURE WORK

In this work, we have investigated GRPO-based reinforcement learning for post-training video MLLMs to enhance their capability in describing actions and events. Our VideoCap-R1 incorporates three key designs: (1) a two-stage generation strategy with structured thinking, (2) Tscore for rewarding the thinking process, and (3) caption score for evaluating the final descriptions. The proposed model demonstrates significant improvements over baseline approaches, achieving superior performance even with limited training samples (1.5k) and outperforming SFT-trained counterparts across all benchmarks. Future work will focus on scaling up the training data to further enhance video description capabilities through reinforcement learning. We hope VideoCap-R1 can serve as a strong foundation for future research on developing more advanced video captioning systems through reinforcement learning techniques.

**Reproducibility statement.** We provide all the necessary details to reproduce our experiments in the Section 4 and Appendix B, C, E, F.

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

# APPENDIX

This supplementary material includes the following sections:

- In Section A, we provide more discussions on structured thinking.

- In Section B, we provide more implementation details.

- In Section C, we provide the details of training data curation.

- In Section D, we provide more qualitative comparisons between VideoCap-R1 and baseline models.

- In Section E, we provide the scoring prompt for Qwen2.5-72B.

- In Section F, we provide the training prompt for Qwen2-VL-7B.

- In Section G, we describe how we use LLMs.

## A  MORE DISCUSSIONS ON STRUCTURED THINKING

**Advantages of structured thinking.** The structured approach enables regex-based extraction of key elements for Tscore evaluation, simplifying the design of the LLM-Free Think Scorer. Our ablation study(Table 4) demonstrates that relying solely on the VLM's inherent reasoning pathways yields limited performance gains. For video captioning, the model primarily needs to perceive and identify core content without engaging in self-reflection to overturn previous recognitions. By rewarding correct structured thinking through Tscore, we effectively promote learning, with experimental results validating this approach.

Table 4: **Ablation study on the thinking process.** Our structured thinking consistently outperforms free-form alternatives.

| Thinking Method | DREAM-1K (F1) | VDC (Acc./Score) | CAREBENCH (Action F1/Object F1) |
|---|---|---|---|
| Baseline | 29.8 | 39.6 / 2.1 | 31.3 / 27.4 |
| a. Free-form thinking | 30.6 | 43.4 / 2.3 | 33.1 / 33.2 |
| b. Guided free-form thinking | 31.4 | 42.5 / 2.3 | 32.6 / 33.5 |
| c. Structured thinking (Ours) | **34.2** | **43.8 / 2.4** | **34.4 / 34.3** |

**Ablation study on the thinking process.** Table 4 shows structured thinking consistently outperforms free-form alternatives. Experimental variations: a. Free-form thinking: Unstructured output without guidance b. Guided free-form thinking: Prompted to consider key elements (subjects/attributes/actions) but unstructured output c. Structured thinking (our method): Formalized output with dedicated thinking reward.

## B    MORE IMPLEMENTATION DETAILS

Table 5 shows the training hyper-parameters in SFT and GRPO. For GRPO optimization, we perform 7 rollouts per prompt($G = 7$) and set the sampling temperature to 1.0. We adopt $\beta = 0.001$ for KL penalization and set the thresholds of event coverage $\delta_1 = 0.28, \delta_2 = 0.35$. All experiments are conducted on 8 H800-80GB GPUs. For GRPO, we allocate 7 GPUs for training and reserve 1 GPU exclusively for rollouts, while SFT utilizes all 8 GPUs for training.

Table 5: Training hyper-parameters of VideoCap-R1.

| Configuration | SFT | GRPO |
|---|---|---|
| Baseline | Qwen2-VL-7B | |
| Optimizer name | AdamW | |
| Optimizer $\beta_1$ | 0.9 | |
| Optimizer $\beta_2$ | 0.999 | |
| Optimizer eps | 1e-6 | 1e-8 |
| Learning rate | 1e-6 | |
| Learning rate schedule | cosine | |
| Training epoch | 1 | |
| Warm-up ratio | 0.05 | 0.01 |
| Weight decay | 0.01 | 0.1 |
| Global batch size | 64 | 56 |

## C    TRAINING DATA CURATION

**Dynamic Video Selection.** We construct our training set by sampling from the Tarsier2-Recap-585K dataset Yuan et al. (2025), as it provides exceptionally accurate and detailed video descriptions with comprehensive action annotations. To ensure the selected videos exhibit sufficient dynamic content for improving action/event description capabilities, we implement an optical-flow-based filtering pipeline that: (1) computes frame-to-frame optical flow intensity as a dynamicity metric, and (2) retains only videos with both high dynamicity scores and appropriate durations (10-30 seconds).

---

**The Format of structured thinking process and final answer**

<think>
#Subject 1: sub1_name [attribute1, attribute2, ...],[action1, action2, ...]
#Subject 2: sub2_name [attribute1, attribute2, ...],[action1, action2, ...]
...
</think>
<answer>
Complete video description
</answer>

---

Figure 4: The Format of structured thinking process and final answer.

**Structured Thinking Annotation.** To effectively reward the model's reasoning process(as shown in Figure 4), we construct specialized training data containing explicit structured thinking annotations. We design a carefully engineered prompt template (Figure 5) to guide Qwen2.5-72B Yang et al. (2024a) in producing structured reasoning content. Each annotation must satisfy: (i) maximal coverage of main subjects while maintaining attribute/action consistency, and (ii) strict temporal alignment between described actions and actual video progression.

**Action-Based Video Filtering.** To ensure balanced action distribution in our training set, we implement an incremental filtering mechanism that maintains a running inventory of covered actions. Videos are selectively added only when they introduce new action types not already represented in our dataset. Through this process, we construct a curated dataset of 1.5K video samples characterized by: (1) high dynamicity, (2) diverse action types, and (3) paired structured reasoning annotations with detailed descriptions.

---

**Prompt Template for Structured Thinking Generation**

Below is a caption of a video: [{caption}]

Extract the main subjects and their corresponding attributes and actions in sequence from the above video description.

**Requirements:**
- "Action" is a verb or phrase.
- All verbs should be in their base form.
- Subject is a single noun that refers to a person or an object.
- Attributes of all subjects cannot be a verb or verb phrase.
- Attributes can be clothes, shoes, hairstyle, belongings for person.
- The main subjects must be clearly defined individuals.
- Each main subject must include at least one action.
- Do not repeat actions.
- Each action must be atomic, meaning it cannot be further divided into multiple actions.
- Changes in subjects due to scene transitions and camera movements are not considered actions.
- Extract the main subjects and their corresponding key actions from the video description, ensuring that each action is listed in order.

Please generate the response in the form of a Python List. DO NOT PROVIDE ANY OTHER OUTPUT TEXT OR EXPLANATION. Only provide the Python List.

For example, your response should look like this:
[ {{ "subject": "subject 1", "attributes": ["attribute1", "attribute2", ...], "actions": [ "action1", "action2", ... ] }},
{{ "subject": "subject 2", "attributes": ["attribute1", "attribute2", ...], "actions": [ "action1", "action2", ... ] }} ]

**Example:**

caption: [In a green field under a clear blue sky, a woman with long hair, wearing a red dress, lifts a smiling child dressed in red and gray above her head. She repeatedly throws the child up and catches her, while the child laughs and enjoys the moment. The woman continues to lift the child high into the air and then brings her back down, repeating this action several times. The child appears joyful and engaged throughout the activity, and the woman maintains a gentle and playful demeanor.]

output:[{{"subject": "Woman", "attributes": ["long hair", "wearing a red dress", "gentle and playful demeanor"], "actions": ["lift a child", "throw up a child", "catch a child", "bring a child back down"]}}, {{"subject": "Child", "attributes": ["dressed in red and gray"], "actions": ["smile"]}}]

---

Figure 5: Prompt Template for Structured Thinking Generation.

## D QUALITATIVE RESULTS

Figures 6–10 present additional qualitative comparisons between VideoCap-R1 and baseline models on representative samples from the DREAM-1K, VDC, and CAREBENCH benchmarks. These examples consistently demonstrate our model's advantages in: (i) fine-grained action characterization, (ii) precise object attribute description, and (iii) comprehensive event coverage across diverse video content.

**Qualitative Case Study**

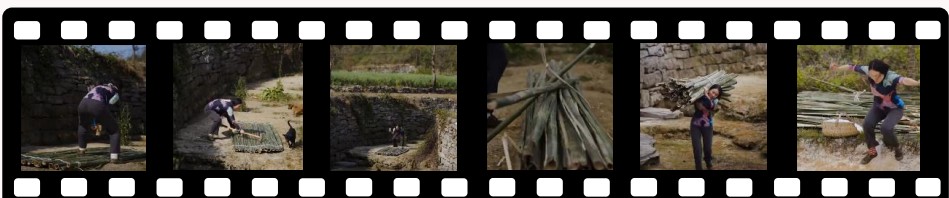

**GPT-4o:** A person is seen working with bamboo poles. They are cutting and splitting the bamboo with an axe. After cutting, they tie the bamboo poles together using rope. The person then lifts the tied bamboo bundle onto their shoulder and carries it away. There are two dogs present in the scene, one black and one brown, which follow the person. The setting is an outdoor area with stone walls and a dry, rocky ground, surrounded by vegetation and fields.

**Qwen2-VL-7B:** The video depicts a woman working in a field, carrying a bundle of bamboo poles on her back. She is seen cutting the poles with an ax and then using them to build a structure. The woman is wearing a blue shirt and black pants, and she appears to be focused on her task. The field is surrounded by trees and other vegetation, and there is a small dog walking around in the background. The woman seems to be skilled at handling the bamboo poles, as she cuts them with ease and uses them to construct a sturdy structure. Overall, the video shows a woman working diligently in a natural setting, using simple tools to create something useful.

**Qwen2-VL-7B-SFT:** The video depicts a rural scene where a woman is engaged in manual labor. She is seen cutting bamboo with a tool, likely an axe or a similar implement, as she works on the ground. The woman is dressed in a traditional outfit, which includes a headscarf and a long-sleeved shirt. She is focused on her task, carefully handling the bamboo to ensure it is cut to the desired length and shape. After finishing her work, the woman carries the cut bamboo on her back, using a traditional method of transporting heavy loads. She is seen walking through a field, with a dog following her closely. The field is surrounded by a stone wall, and there are trees and other vegetation in the background, indicating a rural or agricultural setting.

**VideoCap-R1:**
<think>
#Subject 1: Woman [wearing a colorful shirt, carrying a basket, carrying a bundle of bamboo poles, working in a rural area], [split bamboo poles, carry bamboo poles, carry a basket, jump over a small ditch]
</think>
<answer>
The video captures a rural scene where a woman is working in a field. She is seen using an ax to split bamboo poles on a stone platform. The woman is wearing a colorful shirt and carrying a basket on her back. She is also carrying a bundle of bamboo poles on her shoulders. In the background, there is a stone wall and a field with green crops. The woman is working diligently, splitting the bamboo poles and carrying them to a pile. She then jumps over a small ditch. The video also shows a black dog running around the area, adding a sense of liveliness to the scene. The landscape is surrounded by trees and hills, creating a picturesque rural environment.
</answer>

Figure 6: The first qualitative comparison between VideoCap-R1 and baseline models. The video is from DREAM-1K. We annotate examples with: red underlines for hallucinated content, green underlines for accurate descriptions, and highlighting for correct details uniquely captured by VideoCap-R1 or Qwen2-VL-7B-SFT (vs. Qwen2-VL-7B). Our model demonstrates superior fine-grained content description capabilities.

**Qualitative Case Study**

**GPT-4o:** A man is processing freshly picked green tea leaves outdoors. He holds a large basket filled with tea leaves and pours them into a large wok. He then uses a large spatula to stir and toss the leaves in the wok. Afterward, he pours a large bowl of water into the wok, causing the water to splash and mix with the leaves.

**Qwen2-VL-7B:** The video depicts a man in a white tank top and black shorts standing in front of a large metal pot filled with water. He is holding a large metal basket filled with green tea leaves. The man pours the tea leaves into the pot, and then he uses a large metal spoon to stir the leaves and water together. The scene appears to be taking place outdoors, with a wooden deck and a fence visible in the background. The man seems to be engaged in the process of making tea, and the video captures the moment of the tea leaves being added to the pot. The video is shot in a vertical orientation, and the man is wearing a white tank top and black shorts.

**Qwen2-VL-7B-SFT:** The video depicts a man in a gray tank top and black shorts standing outdoors, holding a large wicker basket filled with green tea leaves. He pours the leaves into a large metal pot, which is then stirred vigorously with a metal shovel. The man continues to stir the pot, and at one point, he adds a bucket of water to the pot, causing the tea leaves to float and swirl. The background shows a wooden deck with a railing and a forested area. The man appears to be engaged in the process of processing the tea leaves, possibly for tea-making purposes

**VideoCap-R1:**
<think>
#Subject 1: Man [wearing a gray tank top, holding a basket of tea leaves], [lift basket, pour tea leaves into a large pot, stir tea leaves with a shovel, add water to the pot]
</think>
<answer>
The video captures a man preparing tea in an outdoor setting. He starts by lifting a large basket filled with fresh tea leaves and pouring them into a large pot. The man then uses a shovel to stir the tea leaves, ensuring they are evenly distributed. After that, he adds a bucket of water to the pot, creating a steaming effect. The background features a wooden deck and a forested area, adding a natural and serene atmosphere to the scene. The man appears to be skilled in the tea-making process, demonstrating his expertise in handling the tea leaves and preparing the pot for further processing.
</answer>

Figure 7: The second qualitative comparison between VideoCap-R1 and baseline models. The video is from DREAM-1K. We annotate examples with: red underlines for hallucinated content, green underlines for accurate descriptions, and highlighting for correct details uniquely captured by VideoCap-R1 or Qwen2-VL-7B-SFT (vs. Qwen2-VL-7B). Our model demonstrates superior fine-grained content description capabilities.

**Qualitative Case Study**

**GPT-4o:** The video captures a playful interaction between a small dog and a crawling baby. The dog closely observes the baby's movements, mimicking the crawling motion and lying down next to the baby. As the baby moves forward, the dog playfully adjusts its position, shifting between lying down and stretching on the carpet. The dog seems curious and highly engaged, creating a heartwarming and entertaining moment of companionship and imitation. The setting is indoors with a calm and comfortable atmosphere, enhancing the cute dynamic between the two.

**Qwen2-VL-7B:** The video shows a baby and a dog playing together on a carpeted floor. The baby is wearing a striped shirt and is lying on the floor on their stomach. The dog is also lying on the floor next to the baby. The baby then begins to crawl towards the dog, and the dog moves closer to the baby. The baby then reaches out and grabs the dog's tail, and the dog rolls over onto its back. The baby then sits up and looks at the camera. The video captures the playful interaction between the baby and the dog, with the baby showing curiosity and the dog responding with playfulness. The carpeted floor provides a comfortable surface for the baby and the dog to play on, and the striped shirt of the baby adds a touch of color to the scene. Overall, the video is a heartwarming display of the bond between humans and animals.

**Qwen2-VL-7B-SFT:** In the video, a baby is seen crawling on the floor, wearing a striped shirt. The baby is positioned in front of a dog, which is also on the floor. The dog appears to be a small breed with a white and brown coat. The baby reaches out to the dog, and the dog moves closer to the baby. The baby then grabs the dog's tail and pulls it, causing the dog to react by moving away. The baby continues to play with the dog, and the dog seems to be enjoying the interaction. The background of the video shows a room with a stone fireplace and a white chair.

**VideoCap-R1:**
<think>
#Subject 1: Baby [wearing striped pajamas, lying on the floor], [crawling, reaching out to the dog, lying down on the floor] #Subject 2: Dog [white and brown, lying on the floor], [sitting on the floor, moving towards the baby, lying down on the floor]
</think>
<answer>
The video captures a heartwarming moment between a baby and a dog in a cozy indoor setting. The baby is lying on the floor, wearing striped pajamas, and appears to be crawling towards the dog. The dog, which is white with brown spots, is sitting on the floor and then moves towards the baby. The baby reaches out to the dog and lies down on the floor, facing the dog. The dog then lies down next to the baby, and the baby looks up at the camera with a smile. The background includes a stone fireplace, a white lamp, and some cables on the floor. The video is a cute depiction of the bond between the baby and the dog, showing their playful interaction.
</answer>

Figure 8: The third qualitative comparison between VideoCap-R1 and baseline models. The video is from CAREBENCH. We annotate examples with: red underlines for hallucinated content, green underlines for accurate descriptions, and highlighting for correct details uniquely captured by VideoCap-R1 or Qwen2-VL-7B-SFT (vs. Qwen2-VL-7B). Our model demonstrates superior fine-grained content description capabilities.

**Qualitative Case Study**

**GPT-4o:** The video showcases a makeup tutorial focused on applying eyeliner. The individual demonstrates a step-by-step process for creating a precise winged eyeliner look, using a liquid eyeliner pen. Throughout the tutorial, they emphasize techniques for achieving symmetry and sharp lines, while adjusting their grip and positioning for optimal control. The background features a soft, pastel color, creating a calming atmosphere as they share tips and tricks to help viewers replicate the eyeliner style easily.

**Qwen2-VL-7B:** The video shows a woman applying eyeliner to her eyelids. She is seen holding a black eyeliner pencil and carefully lining her eyelids. The woman has a light complexion and is wearing a light-colored top. Her hair is pulled back, and she has a small earring in her ear. The background is a light blue color, and there is a watermark in the bottom right corner of the video that reads "Laura Lee Los Angeles." The woman appears to be applying the eyeliner with precision and care, taking her time to ensure that the lines are straight and even. The video is likely a tutorial or demonstration of how to apply eyeliner, as the woman is shown in close-up shots of her face and hands. Overall, the video provides a detailed and informative guide on how to apply eyeliner to achieve a professional and polished look.

**Qwen2-VL-7B-SFT:** In the video, a woman is seen applying eyeliner to her eyelids. She starts by holding a black eyeliner pencil in her right hand and carefully applying it to the inner corner of her eyelid. She then moves the pencil to the outer corner of her eyelid, continuing to apply the eyeliner in a smooth, even line. The woman's nails are painted with a light blue nail polish, and she is wearing a ring on her left hand. The background is a solid light blue color, and the woman is wearing a light-colored top.

**VideoCap-R1:**
<think>
#Subject 1: Woman [with brown hair, wearing a light-colored top, applying eyeliner], [open her eye, hold her eyelid, draw a line with eyeliner, move her head slightly, continue drawing the line, adjust the line]
</think>
<answer>
The video captures a woman applying eyeliner to her eyelids. She is holding her eyelid open with one hand while using a black eyeliner to draw a line along the upper lash line. She is wearing a light-colored top and has brown hair. The background is a solid light blue color. The woman is focused on applying the eyeliner, carefully drawing a straight line from the inner corner of her eye to the outer corner. She adjusts the line slightly to ensure it is even and smooth. Her nails are painted with light blue nail polish, and she is wearing a ring on her finger. The video is a tutorial on how to apply eyeliner, demonstrating the correct technique and providing a close-up view of the process.
</answer>

Figure 9: The fourth qualitative comparison between VideoCap-R1 and baseline models. The video is from CAREBENCH. We annotate examples with: red underlines for hallucinated content, green underlines for accurate descriptions, and highlighting for correct details uniquely captured by VideoCap-R1 or Qwen2-VL-7B-SFT (vs. Qwen2-VL-7B). Our model demonstrates superior fine-grained content description capabilities.

**Qualitative Case Study**

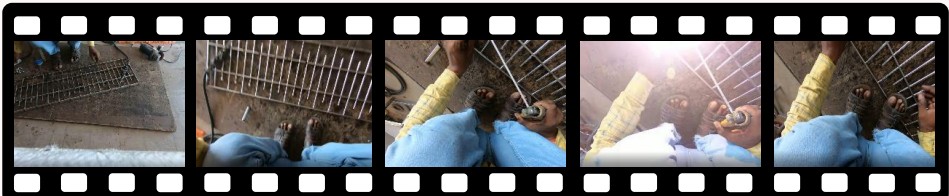

**GPT-4o:** The video shows a person working on a metal grid or frame in a workshop environment. The individual is seen kneeling on the ground while using a welding tool to attach metal pieces together. Throughout the video, various angles focus on the person's hands, feet, and the grid, showcasing the detailed process of welding and assembling. The surroundings include tools and materials typical of a fabrication or metalworking shop. The person wears a yellow plaid shirt and sandals, and there's visible sparks during the welding process.

**Qwen2-VL-7B:** The video shows a man working on a metal grate with a welding tool. He is wearing a yellow shirt and blue jeans, and he appears to be using a welding machine to join the metal pieces together. The man is standing on a wooden floor, and there are various tools and equipment scattered around him, including a hammer, a screwdriver, and a wrench. The scene seems to take place in an industrial or workshop setting, as evidenced by the presence of metal objects and machinery in the background. Overall, the video depicts a skilled craftsman at work, carefully assembling a metal structure using traditional hand tools and modern welding techniques.

**Qwen2-VL-7B-SFT:** The video shows a person working on a metal grid or frame, using a welding tool to join the pieces together. The individual is wearing a yellow shirt and blue jeans, and their feet are visible as they stand on the grid. Various tools and equipment, such as a hammer and a circular saw, are scattered around the work area. The person appears to be focused on the task at hand, carefully manipulating the metal pieces with the welding tool. The overall scene suggests a workshop or construction environment where metalworking is taking place.

**VideoCap-R1:**
<think>
#Subject 1: Worker [wearing a yellow shirt, blue jeans, sandals, holding a welding tool], [adjust metal grid, weld metal grid, move around]
</think>
<answer>
The video captures a worker in a workshop environment, working on a metal grid. The worker is wearing a yellow shirt, blue jeans, and sandals. He is using a welding tool to weld the metal grid, which is placed on a wooden surface. The worker is seen adjusting the grid and welding it in various positions. The workshop is cluttered with tools and equipment, including a grinder and a hose. The worker's movements are focused and deliberate, indicating a skilled and experienced individual. The video provides a close-up view of the welding process, highlighting the worker's attention to detail and the precision required for this task. The overall atmosphere is one of industriousness and craftsmanship, with the worker's hands and the welding tool being the main focus of the scene.
</answer>

Figure 10: The fifth qualitative comparison between VideoCap-R1 and baseline models. The video is from VDC. We annotate examples with: red underlines for hallucinated content, green underlines for accurate descriptions, and highlighting for correct details uniquely captured by VideoCap-R1 or Qwen2-VL-7B-SFT (vs. Qwen2-VL-7B). Our model demonstrates superior fine-grained content description capabilities.

# E    THE SCORING PROMPT FOR QWEN2.5-72B

Figures 11 and 12 present the prompt templates for Qwen2.5-72B to assess caption completeness and naturalness scores, respectively.

---

**Prompt Template for Completeness Score**

Please play the role of a professional video description model evaluation expert. You will be given a video description output by the model, along with the subjects, their attributes, and actions that appear in the video. You will need to rate the completeness and fluency of the video description output by the model on a scale of 1 to 10.

**Video description completeness and fluency rating criteria (1-10 points)**

9-10 points: The description is very complete and fluent, with all subjects (people/animals/objects, etc.) mentioned in the video. The key attributes of the subjects (such as color, size, posture, etc.) are basically covered, and the main actions or interactions of the subjects are accurately described without obvious omissions of important content. The language is clear and logical.

7-8 points: The description is relatively complete but slightly missing. Most of the subjects are mentioned, but some minor subjects may be omitted. The main actions of the subjects are described, but some detailed actions or attributes may be missing; The overall meaning is clear, but there are some vague or incomplete points. Language expression may have flaws or repetitions, but overall it does not affect understanding.

5-6 points: The description is average, with obvious omissions or unclear descriptions of one or more main subjects. The actions or interactions of the subjects are incomplete or inaccurate, and the description is ambiguous or vague, which affects understanding.

3-4 points: The description is severely lacking, only mentioning a small part of the content in the video. The main body and actions are highly summarized or incorrect, and the description cannot help understand the main content of the video.

1-2 points: Almost no descriptive information, description content that is almost unrelated to the video, or completely incorrect, unable to identify the subject, action, or context;

**Video elements**

[{think}]

**Video description**

[{caption}]

**Output Format**

DO NOT PROVIDE ANY OTHER OUTPUT TEXT OR EXPLANATION. Only provide the Python dictionary string. For example, your response should look like this: {{"score": 5}}.

---

Figure 11: Prompt Template for Completeness Score.

# F    THE TRAINING PROMPT FOR QWEN2-VL-7B USED IN GRPO

Figure 13 presents the prompt template employed for both GRPO-based reinforcement learning training and supervised fine-tuning (SFT) on our structured-thinking-augmented instruction dataset.

# G    THE USE OF LARGE LANGUAGE MODELS (LLMS)

We leverage LLMs for data filtering to assist in constructing our training dataset(See Appendix C) and calculating caption score(See Section 3). Beyond these uses, we only employed LLMs for grammar checking during the writing process. The core method development in this research does not involve LLMs as important components.

---

**Prompt Template for Naturalness Score**

**Requirement**

Please play the role of a professional video description model evaluation expert. Please rate the naturalness of the language described in each video based on the following criteria and your own intuitive experience. The rating range is from 1 to 10 points, with higher scores indicating more natural and fluent language.

9-10 points: Extremely natural, logical description, fluent and barrier free language expression, appropriate word choice, natural sentence structure, close to or equivalent to human daily oral or written expression.

7-8 points: relatively natural, overall clear expression, basic logical flow, with a few words or sentence structures that are slightly unnatural, but do not affect understanding.

5-6 points: Generally, there is a certain degree of stiffness in the language, and during the reading process, one can feel that it is not written by humans.

3-4 points: unnatural, with multiple awkward sentences, inappropriate wording, or unclear logic.

1-2 points: Extremely unnatural, clearly perceived as machine language, lacking the fluency and coherence of human language.

**Example**

An example with 9 points: "The video shows a person standing in a doorway, possibly in a house. The person opens the door and steps outside, looking out for a moment before closing the door. The setting appears to be a small, cozy entryway with a doormat in front of the door. The person is wearing a white shirt and is seen moving in and out of the doorway, indicating that they are likely entering or exiting the house. The lighting is warm, suggesting that the video might have been taken in a dimly lit environment, possibly in the evening or at night."

An example with 1 points: "#Subject 1: Person [wearing a sweater], [open the door, step outside, close the door]"

**Video Description**

[{caption}]

**Output Format**

DO NOT PROVIDE ANY OTHER OUTPUT TEXT OR EXPLANATION. Only provide the Python dictionary string. For example, your response should look like this: {{"score": 5}}.

Figure 12: Prompt Template for Naturalness Score.

---

**Prompt Template for Training**

Describe the video in detail using the following process.

**Think about the details in the video:**

Firstly, think how many subjects there are in the video, what attributes each subject has, and then provide the actions/motions sequence for each subject's done. You should put your thoughts into the <think></think> tag in the following json format:

<think>
#Subject 1: sub1_name [attribute1, attribute2, ...], [action1, action2, ...]
#Subject 2: sub2_name [attribute1, attribute2, ...], [action1, action2, ...]
...
</think>

For example:

<think>
#Subject 1: Woman [long hair, wearing a red dress, gentle and playful demeanor], [lift a child, throw up a child, catch a child, bring a child back down]
#Subject 2: Child [dressed in red and gray], [smile]
</think>

**Describe the video:**

Next, you need to use your thoughts to output a complete video description in the <answer></answer> tags. Your description needs to include all the details in think and be organized smoothly and completely.

Figure 13: Prompt Template for Training.

