# OpenReview forum: "VideoCap-R1: Enhancing MLLMs for Video Captioning via Structured Thinking"
_ICLR.cc/2026/Conference — ICLR 2026 Conference Withdrawn Submission_

### Official Review · Reviewer_TBGr · 2025-10-16

**Soundness:** 2
**Presentation:** 3
**Contribution:** 3
**Rating:** 2
**Confidence:** 5

**Summary:**

This paper presents VideoCap-R1, an RL framework based on GRPO for enhancing video captioning in MLLMs. The core idea is to introduce a structured thinking step—where the model first identifies subjects, their attributes, and actions—before generating the final caption. The authors design two specialized reward mechanisms: a LLM-free think scorer that evaluates the quality of the structured reasoning, and a LLM-assisted caption scorer that assesses event coverage and naturalness. Evaluated on three benchmarks (DREAM-1K, VDC, CAREBENCH), VideoCap-R1 significantly outperforms the Qwen2-VL-7B baseline and SFT counterparts using only 1.5K training samples.

**Strengths:**

1. Innovative Structured Reasoning Framework: The two-stage generation process (structured thinking → caption) effectively bridges fine-grained visual perception and fluent description, leading to more accurate and detailed captions—especially for dynamic actions.

2. Effective Reward Design for Open-Ended Tasks: The combination of an LLM-free think scorer (based on attribute/action F1) and an event-based LLM-assisted caption scorer provides robust, objective signals for RL training, successfully adapting GRPO to the challenging open-ended video captioning task.

**Weaknesses:**

1. The absolute performance is not particularly strong: DREAM-1K achieves only an F1 score of 34.2, and VDC attains an accuracy of merely 43.8—metrics that many similarly sized general-purpose models can also reach. In this context, the claimed effectiveness of VideoCap-R1’s caption-specific optimizations is not convincingly demonstrated.

2. The hyperparameters “δ₁ = 0.28, δ₂ = 0.35” appear highly fine-tuned; the paper does not clarify how they were determined. Such precise tuning raises concerns about VideoCap-R1’s robustness and generalizability under different configurations.

3. As a general-purpose R1-like video reasoning model, the authors should have included Video-R1 in their comparisons. Likewise, since VideoCap-R1 is specifically optimized for captioning, it should be benchmarked against VersaVid-R1, another caption-focused video reasoning model.

4. Given that VideoCap-R1 is designed exclusively for captioning, its evaluation is limited to three relatively old benchmarks. It would be valuable to see how it performs on more recent captioning-specific benchmarks, such as VidCapBench or CAPability. (Optional)

**Questions:**

1. The authors employ Qwen2-VL as the base model—an architecture that is already one generation behind (or arguably two, given the recent release of Qwen3-VL). Why not use the more advanced Qwen2.5-VL?
- Moreover, it remains unclear whether VideoCap-R1’s training methodology remains effective when applied to stronger backbones such as Qwen2.5-VL or Qwen3-VL. (Optional)

2. In Equation 6, the two magic numbers lack justification. How were they determined, and what is the underlying objective behind their specific selection?

---

### Official Review · Reviewer_WMzC · 2025-10-30

**Soundness:** 2
**Presentation:** 2
**Contribution:** 3
**Rating:** 6
**Confidence:** 4

**Summary:**

This paper proposes VideoCap-R1, a reinforcement learning (RL) post-training method of multimodal large language models (MLLMs) for video captioning. More specifically, the authors develop VideoCap-R1 by post-train Qwen2VL-7B-Instruct on 1.5k training dataset. They adopt Group Relative Policy Optimization (GRPO) for RL post-training. To apply GRPO for video captioning, this paper proposes a new caption reward model consisting of a LLM-free think scorer and a LLM-assistant caption scorer. This paper evaluates VideoCap-R1 on three benchmarks including DREAM-1K, VDC, and CAREBENCH. The evaluation results show that VideoCap-R1 can provide a higher score compared to existing open-source captioning MLLMs.

**Strengths:**

- S1. [Idea] The basic idea of VideoCap-R1 is to two-stage caption generation that first performs structured reasoning and then synthesizes output captions. This seams intuitive and effective.

- S2. [Solution] The authors successfully applies GRPO to MLLM-based video captioning.

**Weaknesses:**

- W1. [Technical soundness] One of main contributions of this paper is a caption reward design. It consists of a LLM-free thinker scorer (Tscore) and a LLM-assistant caption scorer (CNscore and Escore). However, they seem rather heuristic and do not perform consistently. According to the ablation study in Table 3, Tscore increases the average score by 1.6 compared to the baseline. Also, even though Tscore + Escore provide the highest score, Tscore + CNscore provide a better score on specific benchmarks such as VDC and CAREBENCH.

- W2. [Performance] In Table 2, this paper shows that RL post-training provides a higher score (36.7) than baseline (32.0) and SFT post-training (34.2). However, the improvement does not seem significant.

- W3. [Related work] Regarding the two-stage caption generation, there are some related works in the literature. It would be better to include them in the related work section.
  - [1] Open-Book Video Captioning with Retrieve-Copy-Generate Network. CVPR 2021.
  - [2] Show, Think, and Tell: Thought-Augmented Fine-Tuning of Large Language Models for Video Captioning. CVPR 2024 Workshop.

**Questions:**

- Q1. In Section 3.3, it is mentioned that final reward is Reward = Format_score + Tscore + Escore. What is Format_score?

- Q2. The authors develop a training dataset consisting of 1.5k samples. It seems valuable for the research community. Do the authors plan to make it open-source?

- Q3. Why the authors finally choose the Escore over CNscore? Is it due to the evaluation results? Do the authors have any reasoning for choosing the Escore?

---

### Official Review · Reviewer_Mp3M · 2025-10-31

**Soundness:** 3
**Presentation:** 3
**Contribution:** 3
**Rating:** 6
**Confidence:** 4

**Summary:**

This paper presents VideoCap-R1, a video multimodal large language model trained with GRPO-based reinforcement learning to enhance video captioning. The model first performs structured reasoning to identify subjects, attributes, and actions before producing captions, guided by dual reward functions for structured reasoning and caption quality. Experiments show notable improvements over Qwen2-VL 7B across several benchmarks, demonstrating the effectiveness of reinforcement learning for open-ended video captioning.

**Strengths:**

1. **Method design:** The two-stage framework combining structured reasoning and caption generation is well-motivated, and the use of GRPO-based training with carefully designed reward functions is conceptually sound and aligns naturally with the task.
2. **Strong results:** The proposed model demonstrates consistent and substantial improvements across multiple video captioning benchmarks, validating both the effectiveness and data efficiency of the approach.

**Weaknesses:**

1. **Effectiveness of CNscore:** The ablation results indicate that Escore is generally better than CNscore, making the contribution of introducing CNscore less clear. It remains uncertain what unique benefit CNscore brings given that a more effective alternative already exists.
2. **Overfitting to captioning tasks:** While the method achieves strong performance in video captioning, the design choices, such as structured reasoning tailored for description generation, appear highly specific to this task, potentially limiting the model’s generalization to broader video understanding applications.

**Questions:**

Please refer to Weaknesses. Also, is the final reward a simple addition of Format_score, Tscore, and Escore without coefficients?

---

### Official Review · Reviewer_ZDED · 2025-10-31

**Soundness:** 1
**Presentation:** 3
**Contribution:** 2
**Rating:** 0
**Confidence:** 5

**Summary:**

The paper introduces VideoCap-R1, which is a GRPO-based post-training framework for video captioning. The proposed VideoCap-R1 decomposes captioning into two stages—structured thinking (analyzing subjects, attributes, and actions) and full caption generation—guided by two complementary rewards: a LLM-free think scorer for reasoning quality and a LLM-assisted caption scorer for output completeness and naturalness. Trained on only 1.5k samples, VideoCap-R1 achieves significant improvements over Qwen2-VL-7B and SFT-trained baselines on DREAM-1K, VDC, and CAREBENCH.

**Strengths:**

1. The proposed structured reasoning framework provides a practical RL-based paradigm for video captioning, enabling the model to focus on key subjects, attributes, and actions in videos and thereby generate more comprehensive and accurate captions.

**Weaknesses:**

1. The performance improvement is relatively limited — VideoCap-R1 underperforms Tarsier on the DREAM-1K benchmark and remains notably behind proprietary models such as Gemini-1.5-Pro.
2. The comparison set lacks stronger and more recent baselines (e.g., Gemini-2.5-Pro/Flash, Qwen2.5-VL, InternVL3/3.5), making it difficult to assess competitiveness against state-of-the-art models.
3. The evaluation is confined to only three benchmarks; given that VideoCap-R1 emphasizes structured reasoning for capturing subjects and actions, its effectiveness should also be demonstrated on more targeted datasets such as MotionBench and Favor-Bench to strengthen its empirical validity.

**Questions:**

1. Could the authors provide results on additional benchmarks and include comparisons with more recent models (e.g., Gemini-2.5-Pro/Flash, Qwen2.5-VL, InternVL3/3.5) to better demonstrate the effectiveness and generalizability of the proposed method?

---

### Note · Authors · 2025-11-14

I have read and agree with the venue's withdrawal policy on behalf of myself and my co-authors.